# Effect of Ankle Weights as a Frailty Prevention Strategy in the Community-Dwelling Elderly: A Preliminary Report

**DOI:** 10.3390/ijerph19127350

**Published:** 2022-06-15

**Authors:** Hiroyasu Akatsu, Toshie Manabe, Yoshihiro Kawade, Yoshiyuki Masaki, Shigeru Hoshino, Takashi Jo, Shinya Kobayashi, Tomihiro Hayakawa, Hirotaka Ohara

**Affiliations:** 1Department of Community-Based Medical Education, Nagoya City University Graduate School of Medicine, Nagoya 467-8601, Japan; ykawade@phar.nagoya-cu.ac.jp (Y.K.); hohara@med.nagoya-cu.ac.jp (H.O.); 2Community-Based Integrated Care System Promotion and Research Center, Nagoya City University Hospital, Nagoya 467-0001, Japan; 3Department of Medical Innovation, Nagoya City University Graduate School of Medicine, Nagoya 467-8601, Japan; manabe@kklabo.gr.jp; 4Asuke Hospital, Toyota 444-2351, Japan; y-masaki@juntendo.ac.jp (Y.M.); ezr02623@nifty.ne.jp (S.K.); hayakawatomihiro1031@gmail.com (T.H.); 5Gamagori Municipal Hospital, Gamagori 443-8501, Japan; hoshino-shigeru@city.gamagori.lg.jp (S.H.); jo-takashi@city.gamagori.aichi.jp (T.J.)

**Keywords:** 30 s chair stand test, falls prevention, frailty, resistance training

## Abstract

Since the start of the COVID-19 pandemic, many healthy older adults have been less willing to engage in group exercise for fear of contracting this illness. Therefore, there is a need for an effective home-based exercise program to prevent frailty in the elderly. In this study, we assessed the effectiveness of ankle weights as a frailty prevention device for older adults. The study participants were aged 50–90 years and were screened for falls using the Motor Fitness Scale. Participants were divided into two age groups (≤70 and >70 years) for analysis. Older community-dwelling adults were invited to use ankle weights for 3 months. Seventy-four people responded to the invitation. Physical and cognitive status and performance (body composition, grip strength, standing on one leg with eyes open, the 30 s chair stand test (CS-30), Timed Up and Go test, walking speed, body sway, Japanese version of the Montreal Cognitive Assessment) were assessed before and after 3 months of intervention. CS-30 performance improved during the study. CS-30 reflects lower limb/trunk muscle strength and can be used to indicate the risk of falls. Wearing ankle weights can be recommended for strengthening the muscles of the lower limb and trunk in the elderly.

## 1. Introduction

Prevention of fractures and falls among older adults can help to avoid the need for long-term care, extend healthy life expectancy, and reduce medical and nursing care costs. A variety of exercise interventions are available for community-dwelling older adults to reduce the risk of falls, particularly balance improvement exercises and various types of combination exercises [1,2,3]. Instructor-led gymnastics classes are reportedly safer and more effective than self-directed efforts and self-assessment, and are better able to improve physical function than home-based exercise programs [2,4]. In addition, classes that add nutritional counseling and other services [5] and programs such as the “In Balance program” in the Netherlands [6], which includes an educational component and physical exercises focused on balance and muscle strength, are considered to be more effective. However, widespread implementation of these classes is unlikely in the near future because of systemic shortfalls in facilities, staffing, and funding, and insufficient performance as per the Cost Performance Index. Furthermore, in the context of the coronavirus disease 2019 (COVID-19) pandemic, it has been extremely difficult to conduct calisthenics classes that require physical gatherings of individuals. Staying at home to avoid exposure to COVID-19 could increase the risk of falls and lead to increased frailty in older adults [7]. While advanced efforts for frailty prevention have been reported, including the use of virtual reality [8], we have focused on easily implemented home-based exercises using weights attached to the ankles to improve lower limb muscle strength and maintain balance.

Wearing ankle weights (AWs) while walking has been reported to be an effective method for increasing the intensity of physical activity [9]. AWs are widely available in sporting goods stores, and are often used by younger adults for lower limb training. In one study, oxygen uptake was evaluated using systemic physiological indices, while different levels of load were applied by wearing AWs of different weights [10].

However, the effects of resistance exercise on the lower limbs and the ability of resistance exercise to prevent decline in physical function have not yet been elucidated [11]. Meanwhile, resistance exercise is recommended for older adults, and it has also been suggested that power training may be useful [12]. AWs are used in walking exercise, which requires a certain amount of speed and may be similar to power training. The aims of this prospective study were to assess whether AWs can improve body composition and performance, and to examine the feasibility of AWs for further investigations.

## 2. Materials and Methods

### 2.1. Study Design, Sites, and Participants

We conducted a prospective paired study (with/without feedback) at three sites in Japan, namely the Community Health Education and Research Center (CHC) of Nagoya City University, Asuke Hospital, and Gamagori Municipal Hospital in Aichi. The study participants were healthy volunteers who lived in the towns surrounding the study sites. We recruited subjects using posters and brochures created by the CHC health measurement program. All study participants were required to be aged 50–90 years, have a score of ≥11 (men) or ≥9 (women) on the Motor Fitness Scale [13], be able to respond accurately to questions asked in a consultation with a physician, and be willing to participate. The number of applicants who attended the initial meeting and went on to register after the final interview and screening tests is shown in Appendix A.

The study was approved by the Institutional Review Board of Nagoya City University (approval number 46-18-0006) and registered in the University hospital Medical Information Network (UMIN ID 000038073) on 14 April 2020. All study participants provided written informed consent. The datasets were kept in password-protected systems, and the anonymity of study participants was protected.

### 2.2. Schedule and Recording Daily Activity

The study participants were randomly assigned using an envelope method to attend (group I) or not attend (group II) an individual interview for observing behavioral changes. The effects of the test intervention were analyzed in these two groups separately and together. The study enrollment process is shown in Figure 1.

The flowchart in Figure 1 shows the enrollment procedure and protocol used in this prospective intervention study. After obtaining consent, each participant’s Motor Fitness Scale (MFS) score, and a clinical history, a first set of measurements was obtained. After a 4-week run-in period, a second set of measurements was obtained. After 12 weeks of the intervention, a third and final set of measurements was obtained. This report presents the data for the first and second sets of measurements.

In this study, participants were provided commercially produced AWs (0.5 kg, KW-505; 0.8 and 1 kg, KW-506; 1.5 kg, KW-507; Ironman Club, Taiwan) that they could take home and use freely for the duration of the intervention. The ideal weight of an AW is considered to be 2% of body weight. However, AWs are available only in weights of 0.5 kg, 0.8 kg, 1.0 kg, or 1.5 kg applied to each ankle, so the AWs can only roughly approximate 2% of body weight. Furthermore, each study participant decided the actual weight they would use after wearing the AW for 5–10 min. In addition, activity meters were provided to monitor activity status, but as a result, there were many equipment malfunctions, and these results were not shown in this study.

We set a lower limit of one daily 20 min session of outdoor walking while wearing AWs on at least 2 days per week for 12 weeks. Subjects were free to wear the AWs at any time, without any upper limit on frequency or duration. They could also change the weight of the AWs if they did not feel appropriate to their body weight.

During the study period, subjects were required to record their daily AW use. Individual interviews were conducted with subjects in group I to determine AW-wearing status, identify any physical problems, and provide feedback to increase their motivation (Figure 1, thin arrow).

The first set of physical and fitness measurements was obtained immediately after enrollment (Figure 1, dark arrow), and was followed by a 4-week run-in period (data not reported). A second set of measurements was obtained immediately before the start of the 12 weeks of wearing AWs, and a third set after 12 weeks of wearing AWs. A final set of measurements will be obtained after the AWs have been worn for 1 year. This preliminary report presents the second (pre-intervention) and third sets of measurements.

### 2.3. Lifestyle Questionnaire Survey

Information on the demographic and clinical characteristics of the study participants, including age, sex, occupation, living situation, underlying diseases, medications, smoking and alcohol consumption, and sleep quality, was collected during clinical consultations with a physician using a questionnaire designed by the study investigators. Data on daily activities, participation in calisthenics classes, yoga, walking, dancing, swimming, and participation in sports activities such as tennis, baseball, and golf were also collected.

### 2.4. Muscle Measurements

Body composition parameters, including lower leg circumference and skeletal muscle mass, were assessed using a multi-frequency bioelectrical impedance device (Inbody770; In Body Japan, Tokyo, Japan) [14]. Although there were differences in how this device was used at each facility (as shown in the Appendix A), there were no differences in data acquisition and analysis.

The skeletal muscle mass index (SMI) was derived as the sum of the muscle mass of the four limbs (right arm, left arm, right leg, and left leg) divided by the square of height (kg/m^2^) [15]. Grip strength was reported as a representative indicator [16] and measured using a conventional grip dynamometer (YO2; Tsutsumi Seisakusho Co., Ltd., Chiba, Japan). We hypothesized that muscles related to respiration and swallowing would be strengthened by increasing lower leg muscle strength. Respiratory function was measured during physiological studies at each hospital, and tongue pressure was measured using an Orarize device (JMS, Hiroshima, Japan).

### 2.5. Balance and Mobility Tests

Three measures of balance and mobility (the one-legged stance test (OLST) [17] Timed Up and Go test (TUG) [18], and 30 s chair stand test (CS-30) [19]) were assessed.

The OLST is a balance assessment method used in older adults [17]. In this study, the rater instructed participants to stand on one leg with both upper limbs hanging downwards and eyes open, without specifying any conditions for lifting the other leg. The measurement, with 120 s as the longest measurement time, was conducted twice for each lower limb and the highest value was recorded.

The 3.0 m TUG measures coordination, agility, balance, and speed [18]. The subject starts from a fully seated position with both feet flat on the ground. The subject is then asked to stand up and walk as quickly as possible, without running, around a cone placed 3.0 m in front of a chair and then to return to their initial seated position in the chair. In our study, the shorter time of the two trials was used for the analysis. The TUG was also performed at normal walking speed. A stopwatch was used to record the time of each trial.

The CS-30 measures lower extremity strength [19]. A chair with a seat height of 40 cm is used for the assessment [19]. In our study, the starting position was standardized with regard to buttock placement, back support, use of hands, and foot placement. The participants were asked to cross their arms at the wrists and hold them against the chest. They were then instructed to sit and stand as many times as possible in 30 s. The total number of chair stands completed within 30 s was counted and recorded.

Balance was analyzed using the Gravicorder sway meter for the center of gravity (Anima, Tokyo, Japan). Although Gravicorder device use differed by facility, the acquisition and analysis of the sway of the center of gravity were consistent.

### 2.6. Statistical Analysis

The data are expressed as the median and interquartile range (25th–75th percentile) or as the mean and standard deviation for continuous variables and as the proportion for categorical variables. Participants were divided into two groups (≤70 and >70 years). Because the median age of the study participants was approximately 70 years, the age distributions on either side of this age were similar. Comparisons were made between the variables on the two time points, before and after intervention, with the McNemar test used for categorical variables and the paired t-test for continuous variables. We examined the correlations between age, physical measurement, muscle strength, and balance and mobility tests using Pearson’s correlation coefficients.

Data were analyzed using IBM SPSS software (version 27.0; IBM Corp., Armonk, NY, USA). All analyses were two-tailed, and a *p*-value < 0.05 was considered statistically significant.

## 3. Results

### 3.1. Background Overview and Adverse Events during the Intervention

Ninety-nine people participated in the information session, and consent to participate was obtained from 74 individuals. All participants achieved a passing score on the MFS questionnaire at the initial assessment. One participant withdrew, leaving 73 study participants. There were no further dropouts during the study period. There was one incident in which a study participant lost balance after removing the AWs and sustained a bruise to the face. After consulting a general physician, no problems were identified, and the participant was able to continue with the trial.

General background information about the study participants is shown in Table 1. Overall, 73 people aged 66–76 years (median 71 years) participated, 25% of whom were male and most of whom were homemakers or retirees. Few had ever smoked, and more than half reported having some sleep difficulties. Of the more than 40% of people who had been hospitalized, more than half received medication for an underlying disease (Table 1).

The Pearson coefficients did not reveal any correlations between age, physical measurement, muscle strength, and balance and mobility tests. Furthermore, there were individual differences in participation in sports activities (data not shown in Table 1).

### 3.2. Changes in Body Composition before and after Intervention

Changes in body composition indicated by InBody measurements during the intervention were compared between two groups divided by the median age of 70 years, and are shown in Table 2. There were minor changes in physical composition. SMI was calculated automatically in 44 participants in the CHC and showed a significant decrease after intervention in both the younger and older groups.

### 3.3. Anthropometry

General anthropometry results are presented in the top third of Table 3. There was a significant increase in lower limb circumference in older subjects. There was no change in blood pressure or heart rate.

### 3.4. Performance

Performance measurement data are shown in the middle third of Table 3. In general, the results tend to show slight worsening in younger participants (under 70 years of age) and improvement in those who were older. Tongue pressure tended to increase, but this finding was not statistically significant.

There was a slight increase in grip strength in the younger group, but it was not statistically significant. When asked to walk at their normal speed, participants aged > 70 years walked more slowly in each TUG trial than those aged > 70 years. Examination of individual changes in TUG performance for both walking at normal speed and while walking quickly (Figure 2) revealed that the time taken by younger participants was shorter in many cases.

There was a marked improvement in CS-30 performance, particularly among participants aged > 70 years, with a significantly greater mean number of chair stands performed following the intervention. Examination of the individual data revealed that all participants completed more chair stands in 30 s after the intervention regardless of the number of chair stands before the intervention (Figure 2).

There were no significant changes in cognitive function after the intervention; 10% of participants exhibited mild cognitive impairment, indicated by a MOCA-J test score of ≤26 [20].

### 3.5. Standing Position Balance

The data acquisition and analysis of the sway of the center of gravity using the Gravicorder are shown in the lower third of Table 3. The center-of-gravity sway meter showed no general tendency toward change. Measurements of the front–rear and left–right center of gravity with eyes closed in the younger group were worse after the intervention. Examination of the individual data (not shown) revealed substantial variability among individuals, with some showing improvement after the intervention and others showing no improvement. This variation was more pronounced in older participants.

## 4. Discussion

Multi-component exercise programs that include resistance exercise, balance training, and functional training can be effective in preventing the onset and progression of frailty syndrome in community-dwelling older adults. Furthermore, a combination of balance exercise, functional exercise, and resistance exercise has been reported to reduce the risk of falls [21,22]. Intensified training with trainers [4] or multi-disciplinary teams [23] in addition to multi-component exercise could have optimal effects. Although AWs are used in calisthenics classes for older adults in some areas of Japan, there are various risks involved, and no guidelines currently exist for safe personal use by older individuals. AWs have been found to have a beneficial effect on gait factors when correctly used by healthy adults [24]. However, the effects of simple programs, including resistance training to prevent falls, dancing, and walking, are unknown [1].

Our study targeted older people living in the community who had a relatively low risk of falling, and thus our outcomes were focused on frailty prevention, particularly the effects of muscle strengthening. Effective measures to improve locomotor function among older people include the use of resistance bands [25], iron arrays [26], and machine-based muscle strengthening exercises [27].

In another study, healthy older women who performed lower limb muscle strengthening exercises using a resistance band and wearing AWs three times a week for 12 weeks, including an instruction session once a week, achieved significant improvement in isometric knee extensor muscle strength, isometric elbow flexor muscle strength, grip force, and weight ratio leg extension power but showed no improvement in movement ability, such as standing up and stepping up/down [28]. However, the ongoing COVID-19 pandemic has since led to a decrease in attendance at exercise classes with instructors. In the future, utilization of Internet technology including video, remote instruction, and virtual reality may become important as effective substitutes for face-to-face classes with a trainer [4] or a multi-disciplinary team [23]. In addition, multi-component exercise could help to optimize effects on performance. Although AWs are commonly used in gyms, there is a need to investigate safe and sustainable exercise environments at home. The current study produced primary data confirming the effectiveness of AWs as a wearable muscle loading device. We have sought to contribute to the development of an environment in which exercise can be safely continued at home with AWs. However, depending on the method employed, this approach could cause health problems, and some AW product manufacturers warn against the use of AWs by older adults without supervision. There is currently no consensus on a safe environment for older adults to use AWs as a frailty prevention measure.

In the current study, we implemented a 3-month intervention with minimum requirements for AW use during the intervention period. Data regarding usage frequency and pre-falling incidents obtained from sensors attached to AWs or self-recording were not analyzed against individual anthropometry and performance data. Therefore, this report should be considered an interim component of a larger project. There were no serious accidents or incidents during the study period, and the AW intervention induced significant increases in lower limb circumference and CS-30 performance in older subjects, confirming the beneficial effects of an AW intervention on lower limb muscle strengthening. In particular, the increase in lower limb circumference and improvement in CS30 > 70 (Table 2 and Table 3) indicated that the use of AWs was effective even in people of an older age. Compared with other similar intervention studies [28,29], the focus of the current study was not strict and the sample size was small. Nevertheless, the lower leg circumference of the older participants and the CS-30 performance in both study groups exhibited significant improvement. A previous study reported that the CS-30 test was highly reproducible and significantly correlated with leg extensor muscle strength, and could be used to evaluate lower extremity muscle strength in people aged ≥ 60 years living in the community [19]. In Japan, where the population is aging rapidly, our results suggest that physical status and ability may change at approximately 70 years of age. Figure 2 shows that, although there was no overall improvement in lower extremity muscle strength, there was an average trend toward improvement, and it is possible that differences in individual effort are reflected in the values measured after the intervention. No improvement was observed in other performance items. Some researchers have reported positive effects of AW use on the TUG [30], whereas others have reported negative effects [31]. In future studies, it would be useful to collate each individual’s sensor data and activity diary with these measurements.

Furthermore, in the current study, for additional analysis, we obtained surface electromyography (EMG) and motion capture measurements [32] before and after the intervention in volunteers. These measurements showed increased activity in the proximal portion of the rectus femoris muscle compared with that in the distal portion, as well as increased hip flexion. Although we did not find any significant changes in muscle strength of the lower limbs, trunk muscle mass (Table 2), and tongue pressure (Table 3), future studies should include a longer intervention period and more study participants. The sway of the center of gravity integrates complex functions, including deep sensation and the extrapyramidal tract, and improvement is not only indicated by increased lower limb strength. Previous studies have reported that changes in center of gravity are not directly affected by muscle strength [31,33,34,35].

Considering the attachment site of AWs, the load would be expected to affect the swing motion of the lower limbs and the flexion motion of the hip joint during walking. These movements tend to weaken with aging, and if additional resistance can be selectively applied to these movements, it could not only serve as an exercise load but also suppress the deterioration of walking function among older people. It is also possible that this method could be applied as a high-quality exercise therapy. Because our study regime was not strict, various factors may have affected individual efforts. Furthermore, it was difficult to control for confounding factors, such as the effects of participating in regular individual exercise classes and sports activity, such as yoga, and use of a personal gym.

While staying at home to contain the ongoing COVID-19 pandemic, inactivity among older people has become a serious problem. Frailty prevention approaches are moving toward self-implementation and outdoor activities that avoid close contact. Outdoor activities such as walking while avoiding contact with others are the preferred options for improving physical fitness. Walking has been widely adopted for physical strengthening. However, although walking may have a beneficial effect on cardiopulmonary function, it has been reported to have little effect on muscle strengthening and fall prevention [36]. However, incorporation of walking combined with wearing AWs has the potential to be effective for lower limb muscle strengthening. However, in the case of older adults, because there is a large difference in individual abilities, it is necessary to propose measures that are suitable for each individual’s physical characteristics, muscle mass, muscle strength, and exercise abilities. To provide feedback, a system for formulating a menu that suits each individual according to the guidelines for correct use would be useful.

Finally, one participant remarked that taking part in the study motivated them to exercise and to walk. This comment suggests the importance of fostering and maintaining motivation in healthy older adults. However, this intervention was a relatively minor one in the lives of older individuals and did not have a significant effect. In a previous study conducted in Malaysia, in a 3-month intervention with three types of exercise intensity that were performed for the same duration as in the current study, it was reported that there was no difference between the three groups, although there was an overall effect when exercise intensity was divided into three categories [37].

This study has several limitations. First, because we divided the participants into two groups for comparisons, the sample size for each group was not sufficient. For this reason, the results were not definitive and few measurements were found to differ between the groups. Second, it is difficult to conduct ideal exercise intervention research as a part of the daily lives of older adults. Therefore, it was not possible to collect all the data regarding daily activity and AW-wearing records as planned. Future studies should develop a research system that collects and collates these data automatically. Third, we allowed our study participants to select their AW-wearing conditions themselves and advised a minimum use requirement of 20 min at least once or twice per week. However, some participants found it difficult to understand these instructions.

## 5. Conclusions

This study found significant increases in lower limb circumference and CS-30 performance in older community-dwelling subjects after 3 months of wearing AWs, indicating improved lower limb/trunk muscle strength. This finding suggests that AWs can be recommended as an easy method for strengthening lower limbs in older people. The next step is to develop a research system that can collect and collate study data automatically.

## Figures and Tables

**Figure 1 ijerph-19-07350-f001:**
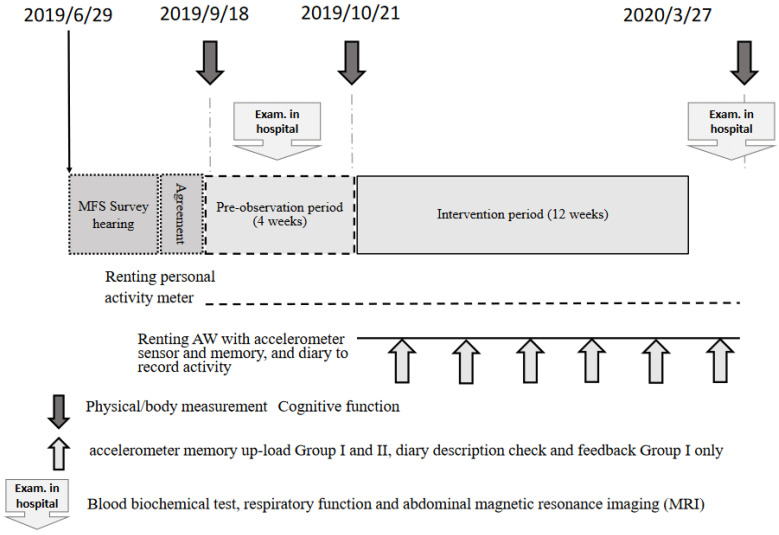
Summary of research schedule.

**Figure 2 ijerph-19-07350-f002:**
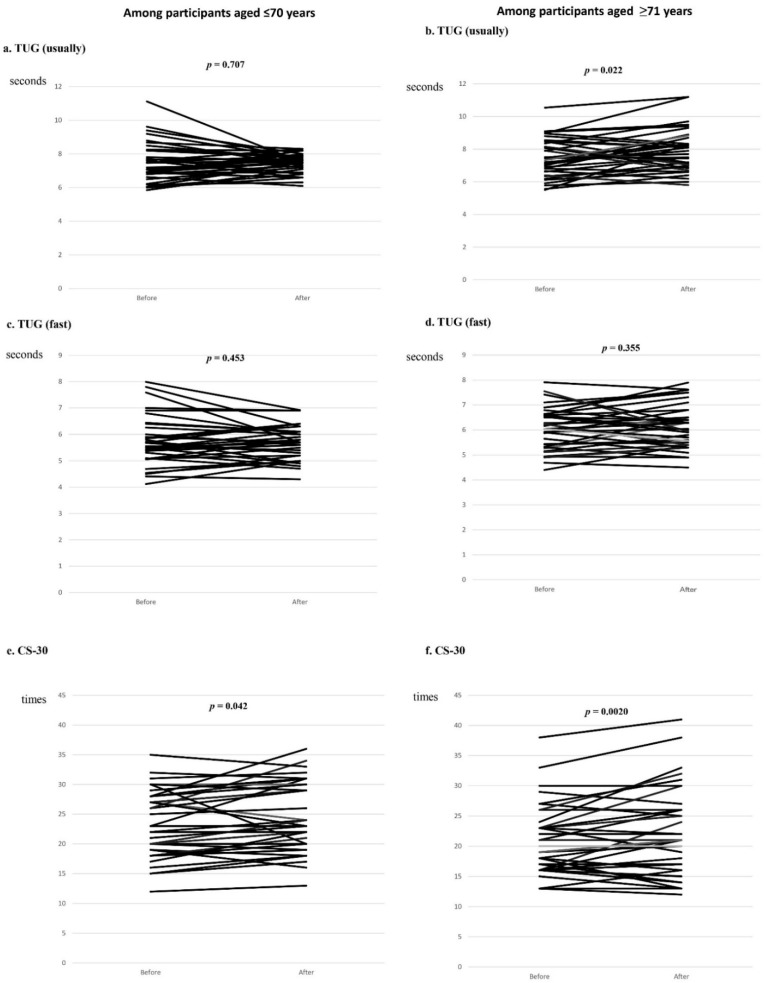
Individual changes in TUG and CS-30 divided around age 70. Graphs showing individual changes in normal walking (**a**,**b**) and rapid walking (**c**,**d**) during the TUG test and the CS-30 scores (**e**,**f**) before and after the intervention for participants according to whether they were aged ≤ 70 years or >70 years. Many participants showed relative improvement in the CS-30. TUG, Timed Up and Go test; CS-30, 30 s chair stand test.

**Table 1 ijerph-19-07350-t001:** Demographic and clinical characteristics of the study participants.

	N = 73
**Demographics**	
Age, years, median (IQR)	71 (66–76)
age > 70, n (%)	37 (50.7)
Men, n (%)	18 (24.7)
Living conditions	
Living with family, n (%)	58 (79.5)
**Employment status**	
Employee	3 (4.1)
Self-employed person	2 (2.7)
Homemaker	27 (37.0)
Retiree	25 (34.2)
Part-time job	11 (15.1)
Other	4 (5.5)
**Lifestyle factors**	
Smoking history, n (%)	6 (8.2)
Alcohol consumption, n (%)	27 (37.0)
Problems with sleep, n (%)	35 (47.9)
**Clinical factors**	
One or more underlying diseases, n (%) (n = 71)	46 (63.0)
Experience of hospitalization, n (%)	31 (42.5)
Taking prescription medication, n (%) (n = 72)	48 (65.8)

**Table 2 ijerph-19-07350-t002:** Body composition of study participants before and after intervention.

	Age, ≤70 Years N = 36			Age, >70 Years N = 37		
Parameters	Before	After	*p*-Value	Before	After	*p*-Value
BMI	23.0 (3.4)	23.0 (3.3)	0.908	22.1 (2.3)	22.2 (2.3)	0.318
Body fat %	29.3 (8.3)	28.9 (8.1)	0.220	28.2 (7.0)	28.1 (7.2)	0.941
Muscle mass	38.8 (7.4)	39.1 (7.8)	0.176	36.0 (7.5)	36.1 (7.2)	0.698
Skeletal muscle mass	22.2 (4.6)	22.4 (4.9)	0.104	20.4 (4.6)	20.5 (4.5)	0.490
Right arm muscle mass	2.0 (0.6)	2.0 (0.5)	0.722	1.8 (0.6)	1.8 (0.6)	0.075
Left arm muscle mass	2.0 (0.6)	2.0 (0.5)	0.865	1.8 (0.5)	1.8 (0.5)	0.097
Trunk muscle mass	18.0 (3.6)	18.0 (3.3)	0.985	16.6 (3.4)	16.7 (3.4)	0.065
Right leg muscle mass	6.4 (1.5)	6.6 (1.7)	0.287	5.8 (1.5)	5.8 (1.5)	0.738
Left leg muscle mass	6.4 (1.5)	6.5 (1.7)	0.310	5.8 (1.5)	5.8 (1.4)	0.596
SMI (n = 22)	6.5 (0.9)	6.4 (0.9)	* 0.045	6.3 (1.0)	6.2 (0.9)	* 0.024

The data are shown as the mean (SD). BMI, body mass index; SMI, skeletal muscle mass index. * Statistically significant at *p* < 0.05.

**Table 3 ijerph-19-07350-t003:** Performance tests and balance in standing position before and after intervention.

	Age, ≤70 Years N = 36			Age, >70 Years N = 37		
Parameters	Before	After	*p*-Value	Before	After	*p*-Value
**Anthropometry**						
Calf circumference						
Right	39.2 (8.2)	40.1 (7.9)	0.551	33.5 (3.0)	33.8 (2.9)	* 0.023
Left	35.1 (2.6)	35.0 (3.1)	0.494	33.6 (3.0)	33.9 (3.1)	* 0.031
Blood pressure (mmHg/mean of right and left arm)						
Systolic	133.3 (23.3)	133.2 (23.0)	0.986	130.1 (18.5)	131.8 (13.0)	0.626
Diastolic	78.4 (16.9)	76.8 (16.3)	0.309	69.6 (10.8)	70.1 (10.4)	0.474
Heart rate(beat/minutes)						
Right	82.0 (13.5)	81.0 (10.8)	0.524	76.8 (9.6)	76.1 (9.0)	0.635
**Performance assessment**						
Tongue pressure	39.2 (8.2)	40.1 (7.9)	0.199	35.6 (8.9)	37.3 (1.4)	0.106
Grip strength (mean of right and left arms)						
Right arm (kg)	30.0 (8.1)	30.2 (7.3)	0.818	25.4 (6.7)	25.4 (6.9)	0.928
Left arm (kg)	27.2 (6.8)	27.5 (6.6)	0.637	25.4 (6.9)	23.3 (6.0)	0.099
OLST total 120, n (%)	21 (58.3)	20 (55.6)	1.000	11 (29.7)	13 (35.1)	0.804
3.0 m walking (second)	1.7 (0.2)	1.6 (0.2)	* 0.012	1.7 (0.3)	1.7 (0.3)	0.225
TUG (usual) (second)	7.5 (1.2)	7.4 (1.2)	0.707	7.5 (1.2)	7.8 (1.3)	(0.022)
TUG (fast) (second)	5.7 (0.9)	5.7 (0.6)	0.453	6.3 (0.8)	6.1 (0.9)	0.355
CS-30 (times)	23.4 (5.6)	24.7 (5.9)	0.042	21.0 (5.9)	22.4 (7.3)	* 0.020
MOCA-J (points)	28.0 (2.1)	28.1 (1.9)	0.782	26.0 (3.0)	26.1 (3.1)	0.772
≤26 points n (%)	32 (88.9)	32 (88.9)	1.000	24 (64.9)	25 (67.6)	1.000
**Balance in standing position**						
**Eyes open**						
Area (cm^2^)	4.48 (1.94)	4.88 (2.67)	0.502	4.29 (2.69)	4.03 (2.16)	0.451
Speed (cm/s)	1.76 (0.47)	1.80 (0.47)	0.655	1.73 (0.48)	1.82 (0.68)	0.186
Density (1/cm)	27.14 (11.02)	27.76 (13.42)	0.820	29.0 (11.37)	30.26 (9.85)	0.401
Center left and right (cm)	0.05 (0.52)	−0.10 (0.66)	0.177	−0.13 (0.90)	−0.04 (0.63)	0.639
Center front and rear (cm)	−0.56 (1.69)	−0.13 (0.90)	* 0.012	−0.13 (0.63)	−0.23 (1.20)	0.706
Berg Balance Scale score	1.33 (0.49)	1.27 (0.49)	0.762	1.33 (0.80)	1.35 (0.49)	0.889
Short area (cm^2^)	10.53 (4.58)	11.79 (6.06)	0.351	10.49 (7.27)	9.03 (5.72)	0.086
Rms value area (cm^2^)	2.39 (1.58)	2.35 (1.36)	0.923	2.00 (5.73)	1.84 (1.08)	0.482
Total track length (cm)	100.29 (36.83)	108.03 (29.24)	0.168	103.56 (28.57)	108.93 (40.63)	0.195
**Eyes closed**						
Area (cm^2^)	5.81(2.87)	5.28 (2.20)	0.177	5.23 (3.43)	5.34 (3.38)	0.823
Speed (cm/s)	2.52 (0.95)	2.30 (0.73)	0.148	2.50 (1.25)	2.52 (1.18)	0.778
Congestion (1/cm)	30.48 (12.93)	29.68 (12.48)	0.652	32.55 (11.59)	32.23 (11.65)	0.870
Center left and right (cm)	0.09 (0.65)	−0.16 (1.78)	* 0.039	−0.12 (0.67)	−0.10 (0.77)	0.935
Center front and rear (cm)	−0.08 (1.78)	−0.82 (1.00)	* 0.021	0.24 (1.61)	0.11 (1.26)	0.650
Berg Balance Scale score	14.58 (7.59)	13.05 (5.09)	0.298	12.97 (8.91)	11.76 (6.78)	0.264
Short area (cm^2^)	2.45 (1.17)	2.25 (0.96)	0.118	2.20 (1.40)	2.20 (1.26)	0.950
Rms value area (cm^2^)	151.23 (56.81)	138.17 (43.83)	0.151	149.71 (75.30)	151.51 (70.53)	0.773

The data are shown as the mean (SD). CS-30, 30 s chair stand test; 3.0 m TUG, timed 2.4 m Up and Go test; OLST, One-leg standing test with eye open. * Statistically significant at *p* < 0.05.

## Data Availability

All relevant data are included in the paper and its Appendix A.

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
