# Peer review of "Effect of Ankle Weights as a Frailty Prevention Strategy in the Community-Dwelling Elderly: A Preliminary Report"

_ijerph, 2022, doi:10.3390/ijerph19127350_

Round 1
Reviewer 1 Report
This study investigated the effect of ankle weights on Physical and cognitive status and performance of older people, and find this exercise intervention is useful to improve older people’s performance in some test. This study is meaningful for this field study. However, the paper needed to be revised in a large extent. The reality of data, the statistical method used and the corresponding results need to be confirmed by authors carefully. The questions are as follows:
- The introduction is too short and only 4 previous studies were citied. This section is needed to enlarge more to explain the necessity of this study sufficiently.
- The reality of data in this study is need to be confirmed, such as in table 1, “employment status” the sum of percentage is 98.6%, not 100%. The same situation also exists in other demographic characters. 3. Which statistical method was used to test the differences of data before and after intervention in group? Such as, table 2 table
- I don’t find the suitable statistical method in the method section. In my opinion, paired T-test should be used.
- In the method section, the author wrote that” Comparisons were made between these two groups using the chi-squared test 160 or Fisher’s exact test for categorical variables and the Mann–Whitney U test or Student’s 161 t-test for continuous variables. W”. however, I don’t find the results about the differences between groups. Also, the author wrote that the correlation and regression method were used, but I also don’t find the relative results in the result section.
- The second and third paragraphs of discussion are too short, suggest to combine them into one paragraph. There are also some short paragraphs in the discussion, suggest to revise them.
Author Response
Professor Dr. Paul B. Tchounwou
Editor-in-Chief
International Journal of Environmental Research and Public Health
27th May, 2022
Dear Professor Tchounwou,
On behalf of my co-authors, I would like to submit a revised version of our manuscript “Effect of ankle weights as a frailty prevention strategy in the community-dwelling elderly: a preliminary report” (Manuscript ID: ijerph-1707705).
We sincerely appreciate the constructive comments and points raised by you and the reviewers. We have carefully considered all of the comments and suggestions, and have revised our manuscript in accordance with each of the points. The revised manuscript text is marked with yellow highlighting, to indicate the changed parts in response to the reviewers’ comments. References have also been added, and thus some of the reference numbers have been changed. The comments have enabled us to substantially improve our manuscript. The manuscript was also resubmitted for proofreading, and changes made in response to comments received are highlighted in light blue. We hope that you will find the revised manuscript suitable for publication in the International Journal of Environmental Research and Public Health.
We have provided point-by-point responses to the comments from the reviewers below. The page and line numbers in the response letter are the ones given in the tracked manuscript specifying . We appreciate your kind consideration of our revised manuscript.
Yours sincerely,
Hiroyasu Akatsu, MD, PhD
Department of Community-based Medical Education
Nagoya City University Graduate School of Medical Sciences
1 Kawasumi, Mizuho-cho, Mizuho-ku, Nagoya 467-8601, Japan
Phone: +81-52-851-5511
E-mail: akatu@med.nagoya-cu.ac.jp
First of all, we have resubmitted the manuscript to the English editing service and have noted below the changes we have made under their suggestion.
(highlighted in light blue)
P1: L15, L43–p2: L46, L47
P3: L95-107
P6: 214-215
P8: L236-237
P10: L245, L249, L259-261
P11: L297, L298, L302-303, L316-317, L323-324
P12: L369
Point-by-Point Responses to the Reviewers’ Comments
(highlighted in yellow)
Reviewer reports:
Reviewer #1:
This study investigated the effect of ankle weights on Physical and cognitive status and performance of older people, and find this exercise intervention is useful to improve older people’s performance in some test. This study is meaningful for this field study. However, the paper needed to be revised in a large extent. The reality of data, the statistical method used and the corresponding results need to be confirmed by authors carefully. The questions are as follows:
Response:
We appreciate the reviewer’s time and valuable feedback. We have carefully considered the comments and suggestions and have revised our manuscript accordingly. We have included our responses to the reviewer’s comments below. For clarity, the reviewer’s comments are shown in blue, and our responses are shown in black.
The revised manuscript text is marked with light blue highlighting, to indicate the changed parts in response to the reviewer’s comments. References have also been added, and thus some of the reference numbers have been changed.
- The introduction is too short and only 4 previous studies were citied. This section is needed to enlarge more to explain the necessity of this study sufficiently.
Response:
We appreciate the reviewer for raising this point. Other reviewers also commented on the length of the Introduction section. Accordingly, we have described additional studies and revised the Introduction section with several reference citations.
P1: L36, L38–41
P2: L49–51, L60-61
As a result, subsequent reference numbers have also been changed.
In addition, ref #20 (L255) has been corrected because of an error in the cited reference.
- The reality of data in this study is need to be confirmed, such as in table 1, “employment status” the sum of percentage is 98.6%, not 100%. The same situation also exists in other demographic characters.
Response:
We appreciate the reviewer for raising this point. We have corrected the percentages for the data on employment status and clinical factors in Table 1.
- Which statistical method was used to test the differences of data before and after intervention in group? Such as, table 2 table
- I don’t find the suitable statistical method in the method section. In my opinion, paired T-test should be used.
In the method section, the author wrote that” Comparisons were made between these two groups using the chi-squared test 160 or Fisher’s exact test for categorical variables and the Mann–Whitney U test or Student’s 161 t-test for continuous variables. W”. however, I don’t find the results about the differences between groups. Also, the author wrote that the correlation and regression method were used, but I also don’t find the relative results in the result section.
Response:
We appreciate these valuable comments. In accordance with the reviewer’s advice, we have rewritten some parts of the statistical analysis section in the Methods section.
P5: L165–187
The data are expressed as the median and interquartile range (25th–75th percentile) or as the mean and standard deviation for continuous variables and as the proportion for categorical variables. Participants were divided into two groups (≤70 and >70 years). The mean age of the study participants was approximately 70 years, and the age distribution either side of this age was similar. Comparisons were made between the variables on the two time points, before and after intervention using the McNemar test for categorical variables or the paired t-test for continuous variables. We examined the correlations between age, physical measurement, muscle strength, and balance and mobility tests using Pearson’s correlation coefficients.
Data were analyzed using IBM SPSS software (version 27.0; IBM Corp., Armonk, NY, USA). All analyses were two-tailed, and a P-value <0.05 was considered statistically significant.
- The second and third paragraphs of discussion are too short, suggest to combine them into one paragraph. There are also some short paragraphs in the discussion, suggest to revise them.
Response:
We appreciate the reviewer for raising this point. The short paragraphs have been revised, in accordance with the reviewer’s suggestion.
P10: L276–280; P11: L308–315, L323–329, L336–342

Reviewer 2 Report
The reviewed article raises an important, current issue of searching for effective forms of training for the elderly that can be performed independently at home. The authors of the article checked the effectiveness of ankle weights as frailty prevention among 73 people (M:18, F:55) aged 66–76.
Strengths:
- An important research problem
Weakness:
- The authors mainly rely on functional tests. Lack of use of qualified research equipment for measuring force and motor coordination.
Question:
- Why do you divide your participants into people over / under 70 years of age? What is the substantive justification for such a division?
Suggestions:
- From my point of view, it should be taken into account what is the daily activity of the respondents. The division into people with high and low daily activity would be interesting. If the Authors do not have such detailed information, I would like to ask for the analysis of the division into professionally active and inactive persons.
- Please add more detailed information about daily activity. In line 181 and in table 1 Authors point out that 20 participants are still professionally active.
Major revision:
- The author compares two age groups <> 70. A suitable test for such analysis is the repeated measures ANOVA.
- In the discussion, the authors do not refer to the differences between the studied age groups.
Need corrections:
- Line 189: “No differences were found between groups I and II”, please add statistical details
- Table 2: Were there any statistically significant differences between the groups in the baseline measurement?
Author Response
Professor Dr. Paul B. Tchounwou
Editor-in-Chief
International Journal of Environmental Research and Public Health
27th May, 2022
Dear Professor Tchounwou,
On behalf of my co-authors, I would like to submit a revised version of our manuscript “Effect of ankle weights as a frailty prevention strategy in the community-dwelling elderly: a preliminary report” (Manuscript ID: ijerph-1707705).
We sincerely appreciate the constructive comments and points raised by you and the reviewers. We have carefully considered all of the comments and suggestions, and have revised our manuscript in accordance with each of the points. The revised manuscript text is marked with yellow highlighting, to indicate the changed parts in response to the reviewers’ comments. References have also been added, and thus some of the reference numbers have been changed. The comments have enabled us to substantially improve our manuscript. The manuscript was also resubmitted for proofreading, and changes made in response to comments received are highlighted in light blue. We hope that you will find the revised manuscript suitable for publication in the International Journal of Environmental Research and Public Health.
We have provided point-by-point responses to the comments from the reviewers below. The page and line numbers in the response letter are the ones given in the tracked manuscript specifying . We appreciate your kind consideration of our revised manuscript.
Yours sincerely,
Hiroyasu Akatsu, MD, PhD
Department of Community-based Medical Education
Nagoya City University Graduate School of Medical Sciences
1 Kawasumi, Mizuho-cho, Mizuho-ku, Nagoya 467-8601, Japan
Phone: +81-52-851-5511
E-mail: akatu@med.nagoya-cu.ac.jp
First of all, we have resubmitted the manuscript to the English editing service and have noted below the changes we have made under their suggestion.
(highlighted in light blue)
P1: L15, L43–p2: L46, L47
P3: L95-107
P6: 214-215
P8: L236-237
P10: L245, L249, L259-261
P11: L297, L298, L302-303, L316-317, L323-324
P12: L369
Point-by-Point Responses to the Reviewers’ Comments
(highlighted in yellow)
Reviewer reports:
Reviewer #2:
The reviewed article raises an important, current issue of searching for effective forms of training for the elderly that can be performed independently at home. The authors of the article checked the effectiveness of ankle weights as frailty prevention among 73 people (M:18, F:55) aged 66–76.
Response:
We appreciate the reviewer’s time and valuable feedback. We have carefully considered the comments and suggestions and have revised our manuscript accordingly. We have included our responses to the reviewer’s comments below. For clarity, the reviewer’s comments are shown in blue, and our responses are shown in black.
The revised manuscript text is marked with light blue highlighting, to indicate the changed parts in response to the reviewer’s comments. References have also been added, and thus some of the reference numbers have been changed.
Strengths:
- An important research problem
Weakness:
- The authors mainly rely on functional tests. Lack of use of qualified research equipment for measuring force and motor coordination.
Response:
We appreciate the reviewer for raising this point. Because the measurements were taken at a local meeting place, it was not possible to perform advanced muscle strength measurements. Grip strength and center of gravity sway tests were the best measurements we were able to perform.
Question:
- Why do you divide your participants into people over / under 70 years of age? What is the substantive justification for such a division?
Response:
We appreciate the reviewer for raising this point. The age distribution of the participants and other factors were used to determine the age groups of the participants. It may be undeniable that the results lack scientific validity.
Suggestions:
- From my point of view, it should be taken into account what is the daily activity of the respondents. The division into people with high and low daily activity would be interesting. If the Authors do not have such detailed information, I would like to ask for the analysis of the division into professionally active and inactive persons.
- Please add more detailed information about daily activity. In line 181 and in table 1 Authors point out that 20 participants are still professionally active.
Response:
We appreciate the reviewer for raising this point. In comparisons of some daily activities, the definitions of these daily activities were fraught with difficulties. We also acknowledge that the amount of activity was picked up from self-recorded data, which is a limitation of the study. This makes it difficult to take into account further information on daily activities.
Major revision:
- The author compares two age groups <> 70. A suitable test for such analysis is the repeated measures ANOVA.
Response:
We appreciate these valuable comments. As mentioned in the response to a comment from Reviewer #1, we found some mistakes in the statistical analysis section in the Methods section. In accordance with the reviewer’s advice, we have rewritten some parts of the statistical analysis section in the Methods section.
P5: L165–187
The data are expressed as the median and interquartile range (25th–75th percentile) or as the mean and standard deviation for continuous variables and as the proportion for categorical variables. Participants were divided into two groups (≤70 and >70 years). The mean age of the study participants was approximately 70 years, and the age distribution either side of this age was similar. Comparisons were made between the variables on the two time points, before and after intervention using the McNemar test for categorical variables or the paired t-test for continuous variables. We examined the correlations between age, physical measurement, muscle strength, and balance and mobility tests using Pearson’s correlation coefficients.
Data were analyzed using IBM SPSS software (version 27.0; IBM Corp., Armonk, NY, USA). All analyses were two-tailed, and a P-value <0.05 was considered statistically significant.
- In the discussion, the authors do not refer to the differences between the studied age groups.
Response:
We appreciate the reviewer for raising this point. The difference has been described on P11: L308–310.
P11: L308–310
In particular, the increase in lower limb circumference and improvement in CS30 in >70 (Table 2 and 3) indicate that the use of AW was effective even in older age.
Need corrections:
- Line 189: “No differences were found between groups I and II”, please add statistical details
- Table 2: Were there any statistically significant differences between the groups in the baseline measurement?
Response:
We appreciate the reviewer for raising this point. Because some unnecessary sentences were mixed, we have deleted these sentences and rewritten the explanation for Table 2.
P6: L207–210
Deleted these sentences.
P7: L212-214
Changes in body composition indicated by InBody measurements during the intervention were compared between two groups divided by the median age of 70 years, and are shown in Table 2.

Reviewer 3 Report
The manuscript is clear and I think that it is useful for professionals dealing with the health and well-being of adult. The references are mostly over the last 5 years, and this aspect affect the study.
The experimental design is appropriate to gain aims of the study and the methods are punctually described, as well as the tables and the figures that properly show the data and make the understanding easy.
The statistical analysis has been well applied. I believe that the article's area of strength are the methodology and the discussion. The limit is the introduction, it’s too poor, the analysis of the literature should be better investigated.
Author Response
Professor Dr. Paul B. Tchounwou
Editor-in-Chief
International Journal of Environmental Research and Public Health
27th May, 2022
Dear Professor Tchounwou,
On behalf of my co-authors, I would like to submit a revised version of our manuscript “Effect of ankle weights as a frailty prevention strategy in the community-dwelling elderly: a preliminary report” (Manuscript ID: ijerph-1707705).
We sincerely appreciate the constructive comments and points raised by you and the reviewers. We have carefully considered all of the comments and suggestions, and have revised our manuscript in accordance with each of the points. The revised manuscript text is marked with yellow highlighting, to indicate the changed parts in response to the reviewers’ comments. References have also been added, and thus some of the reference numbers have been changed. The comments have enabled us to substantially improve our manuscript. The manuscript was also resubmitted for proofreading, and changes made in response to comments received are highlighted in light blue. We hope that you will find the revised manuscript suitable for publication in the International Journal of Environmental Research and Public Health.
We have provided point-by-point responses to the comments from the reviewers below. The page and line numbers in the response letter are the ones given in the tracked manuscript specifying . We appreciate your kind consideration of our revised manuscript.
Yours sincerely,
Hiroyasu Akatsu, MD, PhD
Department of Community-based Medical Education
Nagoya City University Graduate School of Medical Sciences
1 Kawasumi, Mizuho-cho, Mizuho-ku, Nagoya 467-8601, Japan
Phone: +81-52-851-5511
E-mail: akatu@med.nagoya-cu.ac.jp
First of all, we have resubmitted the manuscript to the English editing service and have noted below the changes we have made under their suggestion.
(highlighted in light blue)
P1: L15, L43–p2: L46, L47
P3: L95-107
P6: 214-215
P8: L236-237
P10: L245, L249, L259-261
P11: L297, L298, L302-303, L316-317, L323-324
P12: L369
Point-by-Point Responses to the Reviewers’ Comments
(highlighted in yellow)
Reviewer reports:
Reviewer #3:
The manuscript is clear and I think that it is useful for professionals dealing with the health and well-being of adult. The references are mostly over the last 5 years, and this aspect affect the study.
The experimental design is appropriate to gain aims of the study and the methods are punctually described, as well as the tables and the figures that properly show the data and make the understanding easy.
The statistical analysis has been well applied. I believe that the article's area of strength are the methodology and the discussion. The limit is the introduction, it’s too poor, the analysis of the literature should be better investigated.
Response:
We appreciate the reviewer’s time and valuable feedback. We have carefully considered the comments and suggestions and have revised our manuscript accordingly. We have included our responses to the reviewer’s comments below. For clarity, the reviewer’s comments are shown in blue, and our responses are shown in black.
The revised manuscript text is marked with light blue highlighting, to indicate the changed parts in response to the reviewer’s comments. References have also been added, and thus some of the reference numbers have been changed.
We appreciate the reviewer for raising this point. Other reviewers also commented on the length of the Introduction section. Accordingly, we have cited some additional literature and revised the Introduction section.
P1: L36, L38–41
P2: L49–51, L60-61
As a result, subsequent reference numbers have also been changed.
In addition, ref #20 (L255) has been corrected because of an error in the cited reference.

Round 2
Reviewer 2 Report
The responses to the reviews on the most important issues: the correctness of the analyzes performed and the division into the analyzed groups, do not convince me.
Author Response
First of all, we have resubmitted the manuscript to the English editing service and have noted at light green highlighting sentences in tracked manuscript.
We appreciate the reviewer’s time and valuable feedback. We have carefully considered the comments and suggestions and have revised our manuscript accordingly.
Point-by-Point Responses to the Comments from Reviewer #2
(changes highlighted in light green)
Reviewer reports:
Reviewer #2:
The responses to the reviews on the most important issues: the correctness of the analyzes performed and the division into the analyzed groups, do not convince me.
Response:
The median age of the participants in this study was 70 years. Therefore, we decided to compare the participants in two groups divided by the median age: younger than or equal to 70 years and older than 70 years. The results of the study reflect the general trends in Japan, where the population is aging rapidly, and suggest that physical status and ability may change at approximately 70 years of age.However, a limitation of the study was that the number of samples for each group was not sufficient, and thus the results were not definitive.Based on these comments, we have made some additional alterations to the text, as shown below.
P5: L164–166
Because the median age of the study participants was approximately 70 years, the age distribution on either side of this age were similar.
P11: L314–315
In Japan, where the population is aging rapidly, our results suggest that physical status and ability may change at approximately 70 years of age
P12: L364–366, 370
Because we divided the participants into two groups for comparisons, the sample size for each group was not sufficient. For this reason, the results were not definitive and few measurements were found to differ between the groups.
